# Endophytic Colonization of *Beauveria bassiana* Enhances Drought Stress Tolerance in Tomato via “Water Spender” Pathway

**DOI:** 10.3390/ijms252211949

**Published:** 2024-11-07

**Authors:** Wenbo Guo, Yang Lu, Song Du, Qiyun Li, Xiaowei Zou, Zhengkun Zhang, Li Sui

**Affiliations:** 1Institute of Plant Protection, Jilin Academy of Agricultural Sciences, Gongzhuling 136100, China; guowenbo1999@126.com (W.G.); jluluyang@163.com (Y.L.); ddddsong7386@163.com (S.D.); qyli1225@126.com (Q.L.); zouxiaowei2008@126.com (X.Z.); 2Jilin Key Laboratory of Agricultural Microbiology, Key Laboratory of Integrated Pest Management on Crops in Northeast China, Ministry of Agriculture and Rural Affairs, Gongzhuling 136100, China; 3College of Plant Protection, Henan Agricultural University, Zhengzhou 450002, China; 4College of Agriculture, Jilin University of Agricultural Science and Technology, Jilin 132109, China

**Keywords:** entomogenous fungi, endophytic colonization, tomato, drought tolerance

## Abstract

Drought stress is one of the most important climate-related factors affecting crop production. Tomatoes (*Solanum lycopersicum* L.) are economically important crops which are highly sensitive to drought. The entomopathogenic fungus *Beauveria bassiana*, a widely used biological insecticide, can form symbiotic relationships with plants via endophytic colonization, increasing plant biomass and the ability to resist biotic stress. Under simulated drought stress conditions, the biomass of tomato seedlings such as plant height, root length, stem diameter, fresh weight, and relative water content, as well as the density and size of stomata in tomato leaves were significantly increased after *B. bassiana* colonization via root irrigation (*p* < 0.05). Meanwhile, the physicochemical properties associated with drought resistance such as peroxidase activity and proline content increased significantly (*p* < 0.05), while malondialdehyde reduced significantly (*p* < 0.05), and the expression levels of key genes related to stomatal development and drought tolerance pathways increased significantly (*p* < 0.05). These results indicate that the colonization of *B. bassiana* enhances the water absorption capacity of tomato seedlings and the rate of transpiration significantly and increases drought tolerance in tomato via the “water spender” pathway, which provides a new strategy for improving crop resistance to drought stress.

## 1. Introduction

Drought is one of the most critical abiotic factors directly affecting global crop yields. Tomato production is severely affected by abiotic stress and causes an approximately 60% loss in yields; drought stress in particular causes massive loss [1]. Meanwhile, the increasing demand for water in modern agriculture, shortages in freshwater resources, and continuous population growth are further exacerbating the impact of drought on crop yields, limiting normal plant growth and development [2]. Drought affects a series of physiological activities such as nutrient absorption, photosynthesis, and respiration, thereby affecting the overall growth rate and developmental cycle, which in turn cause a decline in crop yield and quality [3,4]. Effective mitigation of the harm caused by drought stress is therefore an urgent issue in the field of plant protection.

Plants that are sessile, however, have certain strategies to cope with abiotic stress, such as plants that have evolved two major strategies to cope with phosphate limitations (Pi). Interestingly, plant–microbe interactions play an important role in such conditions [5]. Endophytes refer to all microorganisms that can form communities within plants and survive on plant organs, while causing no significant damage to the host [6]. It has also been shown that endophytes can cooperate with plants, reducing the effects of abiotic stress, such as drought via increased tolerance [7]. *Beauveria bassiana* is a broad-spectrum insect pathogenic fungus. In 1991, *B. bassiana* was also identified as an endophytic fungus in plants, possessing the ability to colonize agricultural crops [8]. The mutually beneficial symbiotic relationship between *B. bassiana* and plants has since been revealed [9]; positive effects on plant resistance to biotic stress and biomass have also been shown. For example, artificial inoculation of plants with *B. bassiana* resulted in an effective increase in the mortality rate of insect pests [10]. Colonization of *B. bassiana* via leaf spraying was also found to promote tomato growth and have insecticidal effects against tomato mites [11]. Previous research has also shown that endophytic colonization of *B. bassiana* blastospores induced resistance to various phytopathogens including *sclerotinia sclerotiorum* and *Pythium Myriotylum* [12,13], as well as increased the number of fruits per plant and amount of yield per unit area of tomatoes [14]. However, few reports have examined the effect of *B. bassiana* colonization on plant tolerance to abiotic stress, such as drought.

Long-term evolution has led to the formation about complex mechanisms of stress adaptation and resistance, with various physiological, biochemical, and molecular reactions occurring in plants under drought stress [15,16]. For example, the “drought avoidance” strategy causes changes in the root structure, allowing the stability of the root cell membrane to be maintained and sufficient water to be retained and reducing evapotranspiration via closing of the stomata [17]. Increases in levels of abscisic acid and antioxidant activity also occur, together with the induction of defense mechanisms and the activation of drought stress-related gene expression [18,19]. One of the most common stress resistance strategies in plants is the overproduction of different types of osmolytes, such as proline, mannitol, and sorbitol, which act to promote the stability of reactive oxygen species (ROS) detoxification, the overall structure of the cell membrane, and the natural structure of enzymes [20]. Plants also possess an antioxidant defense system, consisting of superoxide dismutase (SOD) and peroxidase (POD), as well as antioxidants such as carotenoids, all of which work together to protect plants from the destructive effects of ROS [21]. When plants are subjected to drought stress, ROS production and metabolism in plants are disordered. ROS can damage DNA, hinder normal protein synthesis, cause a decrease or even inactivation of enzyme activity in plants, and change a cell membrane’s permeability and structure, resulting in a subsequent content change in malondialdehyde (MDA), which is an important indicator of drought stress in plants. Zhu et al. found that under dehydration and osmotic stress, the overexpression of *StCDPK28* could decrease MDA, its knockdown being contrary to the wild type [22].

Tomato (*Solanum lycopersicum* L.) is one of the most extensively studied vegetable crops as well as one of the most economically important. Drought is a critical abiotic stress, severely restricting tomato growth and yield [23]. Previous study have shown that endophytic colonization of *B. bassiana* in *Zea mays* L. stimulated root growth and flowering, thereby improving drought tolerance [24]. Research has further shown that this endophytic-induced increase in drought tolerance is achieved via two pathways: the “water saver” and “water spender” pathways [25]. However, few reports have examined the effect of the endophytic colonization of *B. bassiana* on drought tolerance in tomatoes, as well as on the mechanisms and pathways involved. In this study, tomato seedlings were colonized with *B. bassiana* blastospores then subjected to drought stress in order to clarify the effect on water utilization pathways and subsequent drought tolerance. The findings provide a new strategy for improving crop resistance to drought stress.

## 2. Results

### 2.1. Colonization of B. bassiana in Tomato Seedlings

The endophytic colonization rate of *B. bassiana* in the tomato seedlings was 74.19%. After being placed in a light incubator for 5–6 days, no white colony was observed in the control treatment, while under *B. bassiana* treatment, white colonies growing around the leaf tissues were observed, the colony morphology being consistent with *B. bassiana* (Appendix A).

### 2.2. Effect of B. bassiana Colonization on Growth of Tomato Seedling Under Simulated Osmotic Stress Conditions in Hydroponics

In the drought stress experiment, no significant differences in growth phenotypes were observed between the control and Bb treatment groups under normal growth conditions. However, after 12 h of drought stress, the differences in growth phenotype gradually became apparent. The leaves of the polyethylene glycol (PEG) treatment group showed slight wilting, curling, and sagging, with significantly greater wilting than the other three treatment groups (Figure 1a). After 24 h of drought stress, leaves of plants in the PEG treatment group showed severe wilting and leaf curling, while drought symptoms were also apparent in the PB group, but to a much lesser degree (Figure 1b). After 36 h of drought stress, all seedlings experienced severe wilting in both the PEG and PB treatment groups (Figure 1c).

### 2.3. Effect of B. bassiana Treatment on Water Absorption

The effect of *B. bassiana* colonization on the water absorption capacity of the tomato seedlings was also determined. On day 4, after being placed in Hoagland nutrient solution, the water absorption capacity of plants in the Bb treatment group was significantly higher than that of the control, at 3.41 ± 0.43 mL and 2.57 ± 0.67 mL, respectively (*p* < 0.05). Meanwhile, on day 8, the water absorption capacity of plants in the Bb treatment group was significantly higher than that of the control (*p* < 0.05), at 8.67 ± 0.91 mL and 6.75 ± 0.95 mL, respectively. Similar findings were also observed on day 12 (*p* < 0.05), at 14.89 ± 1.5 mL and 11.12 ± 2.17 mL, respectively. The water absorption capacity of tomato seedlings increased by 32.68%, 28.44% and 33.90% as compared to the control on day 4, 8, and 12, respectively (Figure 2).

### 2.4. Effect of B. bassiana Colonization on Tomato Seedling Growth Indicators Under Drought Stress in Pot Experiment

The growth phenotype of each treatment group was determined under simulated drought stress conditions in a pot experiment. Under normal growth conditions, the Bb treatment group showed better growth than the control. Meanwhile, on day 1, after drought stress, leaves of plants in the GH treatment group were narrower and wilted compared to those in the GB treatment group. On day 6 of drought stress, seedlings in the GH treatment showed slight wilting, curling, and drooping of leaves, while those in the GB treatment group showed slight drooping. On day 12 of drought stress, seedlings in GH treatment group had dried up and wilted severely compared to those in the GB treatment group, while on day 18, both the GH and GB treatments resulted in severe wilting and withering of the leaves (Figure 3a).

Significant differences in seedling height between treatments were observed on days 1, 6, 12, and 18 after drought stress (*p* < 0.05; Figure 3b). Over time, the seedlings height in groups under drought stress was significantly lower than those of the control and Bb treatments. The height of seedlings in groups treated with *B. bassiana* was higher than that of those treated without *B. bassiana* (Bb vs. control and GB vs. GH). At day 1 and 12 after drought stress, the seedling heights in the Bb and GB treatment groups were significantly higher than those of the control and GH treatment groups, respectively (Figure 3c). Compared to control and GH treatment groups, the height of seedlings in Bb and GB treatment groups increased by 6.20% and 8.42% on day 1, 5.57% and 2.16% on day 6, 5.20% and 4.04% on day 12, and 6.39% and 4.76% on day 18, respectively. These findings indicate that root irrigation with *B. bassiana* caused an increase in the height of the tomato seedlings under both normal and drought stress conditions.

Similar to the results of plant height, significant differences in stem diameter were also observed between treatments on days 1, 6, 12, and 18 after drought stress (*p* < 0.05; Figure 3c). The stem diameter of seedlings treated with *B. bassiana* was higher than that of those treated without *B. bassiana* (Bb vs. control and GB vs. GH) under identical watering conditions, respectively. On days 1, 6, 12, and 18 after drought stress, the stem diameter of seedlings in the GB treatment group was significantly higher than that of plants in the GH treatment group. Meanwhile, the stem diameter was also significantly higher in the Bb treatment compared to the control group. Compared to the control and GH treatment groups, the stem diameter of Bb and GB treatment groups increased by 11.59% and 8.30% on day 1, 13.95% and 7.86% on day 6, 20.12% and 6.20% on day 12, 17.65% and 6.56% on day 18, respectively. These findings indicate that irrigation with *B. bassiana* caused an increase in the stem diameter of tomato seedlings under both normal and drought stress conditions.

Meanwhile, significant differences in root length were observed between treatments on day 18 of drought stress (*p* < 0.05; Figure 3d). The root length in the groups under drought stress was significantly lower than that of the control and Bb treatment groups. Drought caused a significant reduction in root length, although the reduction was smaller in groups treated with *B. bassiana* compared to those treated without *B. bassiana* (Bb vs. control and GB vs. GH) under identical watering conditions, respectively. The root length of Bb and GB treatment groups increased by 35.62% and 27.27% compared to those of the control and GH treatment groups.

There was a significant difference in FW and WC among treatments on day 18 of drought treatment (*p* < 0.05). Both the FW and relative WC of groups treated with *B. bassiana* was higher than that of those treated without *B. bassiana* (Bb vs. control and GB vs. GH) under identical watering conditions, respectively. The fresh weight of the Bb and GB treatment group increased by 17.08% and 48.07% compared to those of the control and GH treatment groups (Figure 3e); likewise, the relative water content of the Bb and GB treatment group increased by 20.54% and 80.75% (Figure 3g). Meanwhile, the DW was lower in the *B. bassiana* treatment groups (Bb and GB) compared to those treated without *B. bassiana* (control and GH). The dry weight of the Bb and GB treatment groups decreased by 16.17% and 37.5% (Figure 3f). These findings indicate that treatment with *B. bassiana* increased the water absorption capacity of the tomato plants.

### 2.5. Effect of B. bassiana Treatment on Stomata Morphology in Tomato Leaves Under Drought Stress

The number of stomata under each treatment was observed and counted during different stages of drought stress using a microscope (Figure 4a). Significant differences in the number of stomata were observed in each treatment on days 1, 6, and 12 of drought stress (*p* < 0.05). Overall, the number of stomata was higher in tomato leaves under drought stress compared to the control and Bb treatment groups, as well as higher in groups treated with *B. bassiana* compared to those groups treated without *B. bassiana* (Bb vs. control and GB vs. GH). Furthermore, with the worsening drought stress, the number of stomata increased significantly under both GH and GB treatment, with a significantly higher number under GB compared to GH treatment. On day 12 of drought stress, the number of stomata per unit area was 34.94 ± 1.51 under GB treatment compared to 30.24 ± 0.69 under GH treatment (*p* < 0.05). The number of stomata per unit area of Bb and GB treatment groups increased by 17.49% and 2.31% compared to those of the control and GH treatment groups (Figure 4c).

Stomatal length and width under each treatment were also observed and measured on day 12 of drought stress using a microscope (Figure 4b). The length and width of stomata were higher in tomato leaves treated with *B. bassiana* compared to those treated without *B. bassiana* (Bb vs. control and GB vs. GH). Stomatal length were greater under Bb and GB treatment (24.53 ± 1.58 μm and 23.57 ± 1.9 μm) compared to the control and GH treatment groups (21.24 ± 1.7 μm and 21.48 ± 1.21 μm), increasing by 15.49% and 9.73%, respectively, as was the case for stomatal widths (16.32 ± 0.96 μm and 16.18 ± 0.7 μm, vs. 15.08 ± 1.42 μm and 15.14 ± 0.76 μm, respectively), which increased by 8.22% and 6.87%, respectively (Figure 4c).

### 2.6. Analysis of Physicochemical Indexes in the Tomato Seedlings

Significant differences in POD activity were observed between treatments on days 1, 6, 12, and 18 of drought stress (*p* < 0.05). Overall, activity was higher in tomato seedlings treated with *B. bassiana* compared to those without (Bb vs. control and GB vs. GH) (*p* < 0.05), suggesting that *B. bassiana* colonization activated POD activity. The content of POD increased by 31.83% and 34.77% on day 1, 9.65% and 18.31% on day 6, 13.75% and 14.54% on day 12, 16.62% and 0.33% on day 18, respectively (Bb vs. control, GB vs. GH) (Figure 5a).

Significant increases in PRO content were also observed under GH and GB treatment compared to Bb and control treatment on days 1, 6, 12, and 18 of drought stress (*p* < 0.05). These findings indicate that both drought and *B. bassiana* treatment result in an increase with the PRO content. Compared with GH treatment, a significant increase with PRO content was observed under GB treatment on days 1, 6, 12, and 18 of drought stress (*p* < 0.05); the content of PRO was increased by 16.32% and 18.83% on day 1, 60.09% and 17.18% on day 6, 121.32% and 88.30% on day 12, 61.33% and 2.60% on day 18, respectively (Bb vs. control, GB vs. GH), suggesting that under drought conditions, *B. bassiana* treatment caused a further increase in PRO content in the tomato seedlings (Figure 5b).

Compared to the control treatment, a decrease in MDA content was observed on day 1 of drought stress in plants treated with *B. bassiana* (*p* < 0.05). Overall, drought stress caused a significant increase in MDA accumulation in all plants (*p* < 0.05); however, a significant reduction was observed under GB compared to GH treatment. This indicates that treatment with *B. bassiana* inhibited the drought stress-induced accumulation of MDA in the tomato seedlings. The MDA content of Bb and GB treatment groups decreased by 14.08% and 21.81% on day 1, 15.99% and 43.24% on day 6, 11.48% and 42.10% on day 12, and 12.88% and 36.25% on day 18, respectively (Bb vs. control, GB vs. GH) (Figure 5c).

### 2.7. Analysis of Gene Expression Levels in the Tomato Seedlings

Compared with the control, three genes related to signaling pathways involved in plant stomatal development and regulation (*Solyc08g061560.2*, *Solyc12g042760.1,* and *Solyc09g091760.1*) were upregulated on day 1 of drought stress following treatment with *B. bassiana* (Bb and GB) (Figure 6a–c). On day 12 of drought treatment, the expression of all three genes was upregulated under both *B. bassiana* treatment and drought stress (Bb, GH, and GB). In addition, on day 1 of drought stress, treatment with *B. bassiana* also resulted in the upregulation of relative expression levels of five drought resistance pathway-related genes (Figure 6d–h). On day 12 of drought stress, all genes except for the *PSY* gene were upregulated in the Bb treatment group compared to the control. These findings indicate that treatment with *B. bassiana* induced the expression of drought resistance-related genes in the tomato seedlings.

## 3. Discussion

Numerous studies have focused on the problem of drought in crop production; for example, in terms of the management of water conservancy facilities [26,27], breeding of drought tolerant varieties [28], use of mulching [29], moisture preservation [30], and the addition of exogenous substances to enhance plant drought tolerance [31,32]. Exogenous application of 2,4-electrobrassinolide (EBR) was found to effectively alleviate the wilting of tomato seedlings and the decrease in the relative water content of leaves [33], while increasing the activity of SOD, POD, and CAT enzymes, and the expression of related genes (FeSOD, POD, and CAT). Meanwhile, in tobacco, increases in osmoregulatory substances such as proline and soluble sugars and the content of plant growth hormone (IAA) and brassinolide, as well as upregulation of drought tolerance-related genes induced by BR (*BRL3* and *BZR2*) and IAA (*YUCCA6*, *SAUR32*, and *IAA26*) signaling pathways, were also observed, all of which served to enhance drought resistance [34,35,36]. Studies have also shown that endophytic microorganisms have a significant positive effect on plant drought tolerance [37,38,39]. *B. bassiana* is an important biological insecticide, and has been widely commercialized for the biological control of various agricultural pests [40,41,42]. Recent studies have further shown that *B. bassiana* can colonize plants and alter their biological characteristics, promoting growth as well as disease and insect resistance [43,44,45,46,47]. Our previous results indicated that compared to hydrophobic conidia, hydrophilic blastospores of *B. bassiana* not only resulted in higher resistance to *Botrytis cinerea* infection after endophytic colonization in tomatoes [48], but also improved tomato yield [14]. In the present study, we found that colonization of *B. bassiana* blastospores significantly improved drought tolerance in tomato seedlings under both potted drought stress and hydroponic PEG-simulated drought conditions. Compared to hydrophobic conidia, hydrophilic blastospores offer various advantages because of their degree of water solubility and ease of production.

A previous study found that colonization of *B. bassiana* promoted the growth of maize roots, thereby improving their tolerance to drought stress [44]. In this study, the endophytic colonization of *B. bassiana* promoted the root length and water absorption capacity of tomato seedlings under drought stress. It has already been suggested and studied that plants enhance their tolerance to drought stress by changing their morphological structure, for example, by increasing their root to shoot ratio, decreasing stomatal aperture size, and increasing stomatal density, all of which are common response mechanisms during plant adaptation to drought stress [49]. A previous study revealed that colonization with different strains of *B. bassiana* resulted in a significant increase in the number of tobacco epidermal hairs, the stomatal density, and the stomatal index, thereby promoting the development of tobacco epidermal stomata [50]. Meanwhile, an increase in leaf stomatal density was also observed in rice and Leymus chinensis plants under mild and moderate drought stress, although a decrease in leaf stomatal density was observed under severe drought conditions [51]. The results showed that the colonization of *B. bassiana* in tomato seedlings caused an increase in the expression of key genes related to stomatal development, and led to a significant increase in stomatal density and size, which affected water transpiration. Aimone et al. found that colonization by endophytic fungi improves drought tolerance via two main pathways: the “water saver” and “water spender” pathways [25]. The results of this study showed that the endophytic colonization of *B. bassiana* in tomato promoted the water absorption of plant roots and maintained the stability of the plant cell membrane, thereby enhancing the drought resistance of tomato.

Plants have developed various mechanisms of drought tolerance. Of these, the clearance of free radicals (ROS) plays an important role in short-term survival under severe stress, while metabolic change contributes to long-term protection [52]. Under drought conditions, plants regulate their enzymatic system via superoxide dismutase (SOD), peroxidase (POD), catalase (CAT), glutathione reductase (GR), and other enzymes that act to clear ROS. Of these, POD is known to cooperate with SOD to eliminate ROS, thereby enhancing plant resistance. In this study, colonization of *B. bassiana* under drought stress significantly increased POD activity in tomato seedlings, resulting in the effective removal of ROS and subsequent protection from or a reduced risk of drought-induced damage. Osmotic adjustment (OA) has been shown to be an important physiological adaptation to drought stress, improving overall drought tolerance [15].

Proline dehydrogenase (PRO) is a highly water-soluble substance that helps maintain the water content of plant cells and tissues [53,54]. An increase in free proline was found to not only alleviate damage caused by drought-induced water loss in plants, but also maintain water retention in lemon leaves’ cells and tissues [55,56]. In this study, even under normal growth conditions, *B. bassiana* colonization caused a significant increase in the content of proline in the tomato seedlings. Furthermore, under drought stress, an increase in proline was observed under all treatments, although this was significantly higher in plants treated with *B. bassiana*. PRO plays an important role in the synthesis of proline, and in this study, a significant increase in PRO activity was observed in tomato seedlings treated with *B. bassiana*. Meanwhile, in contrast, MDA disrupts the structure and function of proteins, thereby disrupting the normal physiological and metabolic rhythm of plants, preventing healthy growth and development. MDA plays an important role in plant membrane lipid peroxidation, changes in which can serve as an important indicator of stress-induced damage. Research has also shown a significant positive correlation between changes in proline content and MDA in plants under stress. For example, under combined drought and salt stress, the membrane permeability of rice seedlings was found to increase together with increases in contents of proline and MDA [57]. In this study, the MDA content of the tomato seedlings increased significantly following drought stress; however, a significant decrease was observed in plants colonized with *B. bassiana*. These findings suggest that *B. bassiana* colonization significantly reduced the degree of membrane lipid peroxidation in tomato plants under drought stress.

A previous study revealed that the colonization of *B. bassiana* caused an increase in resistance to biotic stress and significant increase in expression levels of key genes related to plant resistance pathways, such as the salicylic acid pathway, jasmonic acid (JA) pathway, and ethylene pathway [50]. *B. bassiana* colonization was also found to induce the upregulation of key genes related to these pathways in tomatoes, thereby increasing resistance to *B. cinerea* and *Sclerotinia sclerotiorum* [13,14]. Plant tolerance to drought is known to be closely related to various physiological pathways, including the JA pathway, ascorbate peroxidase (APX) pathway, nitrogen metabolism pathway, carotenoid biosynthesis pathway, and ABA biosynthesis pathway [58]. When subjected to drought stress, the expression levels of associated genes are upregulated, thereby activating the above-mentioned pathways. In line with this, the endophytic colonization of *B. bassiana* significantly increased the expression levels of key genes in the tomato seedlings under drought stress, thereby activating related pathways; nevertheless, the biochemical levels of chemicals such as endogenous JA and nitrogen need to be identified in further studies (Figure 7).

## 4. Materials and Methods

The tomato species BEAUTY was bred by Jilin Dalu Seed Industry Co., Ltd. (Gongzhuling, China). Peat soil was obtained from Pindstrup Mosebrug A/S (Ryomgard, Denmark) and used for the drought stress pot experiments. *B. bassiana* strain D1–5 [14] was isolated and identified from diseased larvae of *Ostrinia furnacalis* in the field by the Institute of Plant Protection, Jilin Academy of Agricultural Sciences, Changchun, China.

*B. bassiana* conidia were scraped off then cultured in sterilized Sabouraud Dextrose Broth with Yeast Extract (SDY) medium for 96 h in a shaker at 26 °C and 200 r/min. They were then centrifuged and filtered to obtain clean blastospores [59]. The suspension was diluted using an appropriate amount of 0.05% (*v*/*v*) Tween-80 then mixed evenly to a final concentration of 1 × 10^8^ spores/mL, as determined using a hemocytometer under a microscope. The samples were then stored at 4 °C until use [48]. Healthy tomato seeds of uniform size were selected for inoculation. The plant surface were disinfected with 1% sodium hypochlorite (NaClO) before sowing in pots (height: 18.5 cm, diameter: 15.2 cm) containing peat soil. When seedlings reached a height of 10 cm, root irrigation with *B. bassiana* blastospores was carried out. As the control, seedlings were irrigated with 0.05% (*v*/*v*) Tween-80. A total of three replicates were prepared with 10 plants per replicate. The inoculation amount per plant was 40 mL, with a 24 h interval between each treatment. To calculate the colonization rate of *B. bassiana*, we used the colonization rate equation:$$\mathrm{Colonization}\;\mathrm{rate}\;\left(\%\right)=\frac{\mathrm{The}\;\mathrm{number}\;\mathrm{of}\;B.\;bassiana\;\mathrm{colonized}\;\mathrm{plants}}{\mathrm{Total}\;\mathrm{number}\;\mathrm{of}\;\mathrm{plants}}\times 100$$

PEG treatment was employed to induce drought stress under hydroponic conditions [60]. Briefly, tomato seedlings (BBCH 30–39) irrigated with *B. bassiana* or 0.05% (*v*/*v*) Tween-80 and showing consistent growth were held under running water to remove any soil from their roots. They were then washed three times with distilled water and placed in 500 mL triangular bottles containing Hoagland nutrient solution for 12 h to eliminate damage caused during transplantation.

Four treatments were examined in this experiment. Treatment 1: tomato seedlings root irrigated with 0.05% (*v*/*v*) Tween-80 then placed in Hoagland nutrient solution (control); Treatment 2: tomato seedlings root irrigated with *B. bassiana* then placed in Hoagland nutrient solution (Bb); Treatment 3: tomato seedlings root irrigated with 0.05% (*v*/*v*) Tween-80 then placed in Hoagland nutrient solution with a final concentration of 8% (*w*/*v*) PEG-6000 (PEG); Treatment 4: tomato seedlings root irrigated with *B. bassiana* then placed in Hoagland nutrient solution with a final concentration of 8% PEG-6000 (PB). Morphological changes in the tomato seedlings were observed and recorded at 0, 12, 24, and 36 h after treatment, and photos were obtained by an electronic camera. Each treatment involved 10 plants, all of which were placed in a greenhouse at a constant temperature of 26 °C.

A hydroponics experiment was conducted to determine the effects of *B. bassiana* colonization on the water absorption ability of the tomato seedlings (BBCH 30–39). Tomato seedlings irrigated with either 0.05% (*v*/*v*) Tween-80 or *B. bassiana* and showing consistent growth were selected. Rhizosphere soil was removed as described above, then all the seedlings were placed in tubes containing 50 mL Hoagland nutrient solution for 12 days. Two treatments were examined in this experiment, with 10 seedlings per treatment. Treatment 1: tomato seedlings root irrigated with 0.05% (*v*/*v*) Tween-80 root then placed in Hoagland nutrient solution (control); Treatment 2: tomato seedlings root irrigated with *B. bassiana* then placed in Hoagland nutrient solution (Bb). The amount of water remaining in each tube was observed and recorded every 2 days to determine the rate of absorption [61].

A pot experiment was carried out to examine the effect of drought stress on tomato seedling biomass (BBCH 30–39). Four treatments were examined, with three replicates of 10 seedlings per treatment. Treatment 1: tomato seedlings root irrigated with 0.05% (*v*/*v*) Tween-80 root irrigation (control); Treatment 2: tomato seedlings root irrigated with *B. bassiana* (Bb); Treatment 3: tomato seedlings root irrigated with 0.05% (*v*/*v*) Tween-80 then exposed to natural drought (GH); Treatment 4: tomato seedlings root irrigated with *B. bassiana* then exposed to natural drought (GB). Under treatments 1 and 2, plants root irrigated with Tween-80 or *B. bassiana* were irrigated with an equal amount of distilled water three days after treatment and every 3 days thereafter, while no water was provided under treatments 3 and 4. The evaluation of plant drought stress was determined based on whether the surface soil was dry in the pots. Height and stem diameters were recorded and measured 1, 6, 12, and 18 days after observing drought stress, while root length, fresh weight (FW), and dry weight (DW) were measured on day 18. The length of the seedling from the soil level to the highest point of growth was determined using a ruler as a measure of plant height, while the diameter of the stem one-third below the cotyledon was determined using a vernier caliper as a measure of stem diameter. To determine root length, underground parts were removed and washed, then surface moisture was absorbed before measuring the length using a vernier caliper. FW was determined using a TP-114 thousandth analytical balance (Beijing Sadolis Instrument System Co., Ltd., Beijing, China), while DW was determined after drying. Briefly, whole seedlings with soil removed were placed in a 105 °C oven for 30 min to induce withering, then in an oven set to 65 °C for 24 h. The final weight was determined as a measure of DW [62]. The relative water content (WC) of the plants was then calculated as follows:$$\mathrm{WC}\;\left(\%\right)=\frac{\mathrm{FD}-\mathrm{FW}}{\mathrm{FW}}\times 100$$

On day 12 of natural drought stress, 10 tomato seedlings from the drought stress pot experiment were randomly selected from each of the three replicates in each treatment group. Samples were then taken from the same position of the third fully expanded leaf from the top in each seedling. Dust and dirt on the leaf surface were gently removed using sterile cotton balls, then a thin layer of nail polish was applied on the front of the leaf. The leaf surface was then allowed to dry naturally for 10–20 min. After drying, the nail polish was removed from the leaf, spread onto a slide and covered. A Leica SP8 laser scanning confocal microscope (Leica Microsystems (Shanghai) Trading Co., Ltd., Shanghai, China) was then used to obtain images of the stomata under five different fields of view. The number of stomata in each area was determined under 20× magnification, then one stoma from three images in each replicate was randomly selected for measurement of the length and width at 40× magnification [63].

Leaves were collected from each of the four treatment groups in the drought stress pot experiments for analysis of physicochemical properties and gene expression levels. Five randomly selected seedlings from each replicate were sampled on days 1, 6, 12, and 18 of drought stress. The top three leaves were then removed and stored at −80 °C until use. A Peroxidase assay kit (A084-3-1), Proline detection kit (A107-1-1), and Plant Malondialdehyde Detection Kit (A003-3-1) (Nanjing Jiancheng Biotechnology Company, Nanjing, China) were used for the analysis of POD, PRO, and MDA contents, respectively, based on the manufacturers’ instructions.

Analysis of the following stress resistance-related genes was also carried out: the *AOS* gene, which is involved in jasmonic acid (JA) synthesis and the signal response pathway [64]; the *APX2* gene, which is related to ascorbate peroxidase and redox homeostasis [65]; the *NIR1* gene, which is associated with nitrite reductase related to the nitrogen metabolism pathway [66]; the *PSY* gene, which is associated with the limiting enzyme of carotenoid and abscisic acid (ABA) biosynthesis [67]; and the *FAB2* gene, which is associated with fatty acid (FA) biosynthesis [68]. Total RNA was extracted from the tomato leaf samples using TRIzol reagent (Nanjing Jiancheng Biotechnology Company, Nanjing, China) according to the manufacturer’s instructions, then reverse transcription was performed using the total RNA as a template for synthesis of cDNA [69]. Real-time quantitative RT-PCR(qPCR) was then performed to determine gene expression levels under different treatments using the 2-ΔΔCT relative quantification method [70]. The mRNA sequences of the eight candidate genes were determined in the NCBI database (https://www.ncbi.nlm.nih.gov/) (*Solyc08g061560.2*, *Solyc12g042760.1* and *Solyc09g091760.1* were accessed on 24 October 2024; *AOS*(834273), *APX2*(820121), *NIR1*(816055), *PSY*(831587) and *FAB2*(818973) were accessed on 18 September 2024) [71,72], and the Actin 7 gene (ACT) was used as a reference [73]. The specific primers, sample cDNA synthesis reaction system, and qPCR conditions are listed in Appendix A. The primers were synthesized by Sangong Biotechnology Co., Ltd. (Shanghai, China).

All experimental results represent the average of all replicates. Multiple comparison tests were performed on all datasets using IBM SPSS Statistics 26 software to evaluate te effects of *B. bassiana* colonization under drought stress on the root growth, biomass, stomatal changes, enzyme activity, and drought-related gene expression of tomato seedlings. We used one-way ANOVA to test the significant difference, and *p* values were estimated via Brown–Forsythe and Barlett’s tests (significant differences were considered at *p* ≤ 0.05). GraphPad 9.0 software was used to plot the results. As compared to the control, the percentage of effects was checked using the following formula [74].
$$\mathrm{Percentage}\left(\%\right)=\frac{\mathrm{treatment}-\mathrm{control}}{\mathrm{control}}\times 100$$

## 5. Conclusions

The results showed that after *B. bassiana* colonization, the plant height, root length, stem diameter, and root water absorption of tomato seedlings were significantly increased under drought stress, which enhanced the health and water absorption capacity of plants. The activity of the POD enzyme and the content of PRO in tomato seedlings were significantly increased (*p* < 0.05), while the content of MDA was significantly decreased (*p* < 0.05), which protected plants against osmotic stress under drought stress. Meanwhile, the expression level of key genes related to stomatal regulation and drought tolerance were also significantly upregulated (*p* < 0.05). These results suggested that the colonization of tomato seedlings by *B. bassiana* enhanced tomato’s drought resistance through the “water spender” pathway (Figure 7). Overall, these findings provide new strategies for improving tomato plant survivability under drought stress.

## Figures and Tables

**Figure 1 ijms-25-11949-f001:**
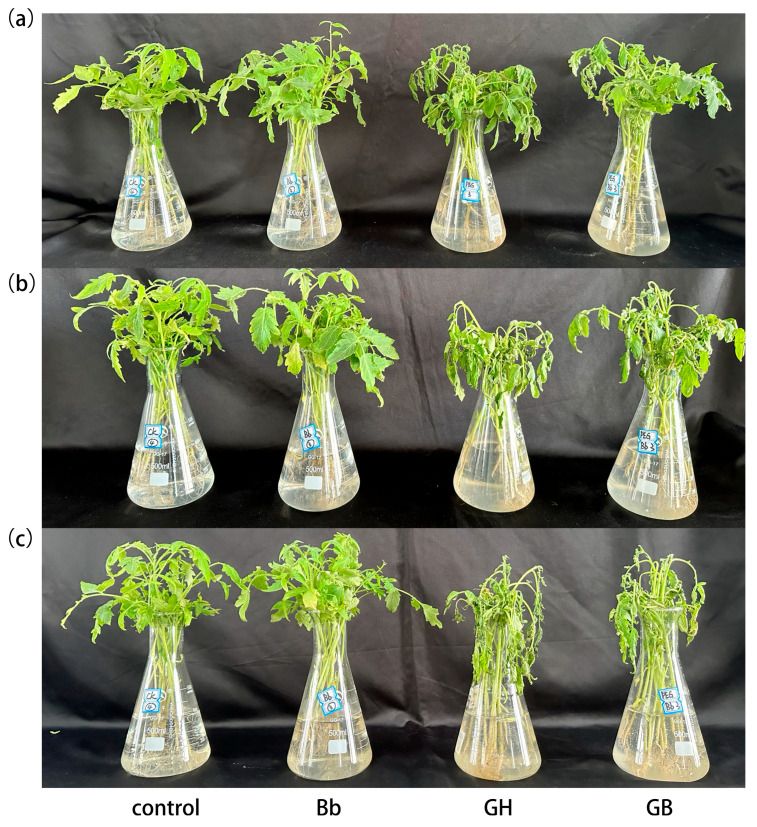
Effect of *B. bassiana* colonization on phenotype of hydroponically grown tomato seedlings at different stages of drought stress. Seedlings treated with (**a**) PEG-6000 for 12 h, (**b**) PEG-6000 for 24 h, and (**c**) PEG-6000 for 36 h. From left to right are control, Bb, GH, and GB treatments. Control seedlings were placed in Hoagland nutrient solution after root irrigation with 0.05% (*v*/*v*) Tween-80. Bb seedlings were placed in Hoagland nutrient solution after root irrigation with *B. bassiana*. PEG seedlings were placed in Hoagland nutrient solution containing 8%PEG-6000 after root irrigation with 0.05% (*v*/*v*) Tween-80 irrigation. PB seedlings were placed in Hoagland nutrient solution containing 8%PEG-6000 after root irrigation with *B. bassiana*.

**Figure 2 ijms-25-11949-f002:**
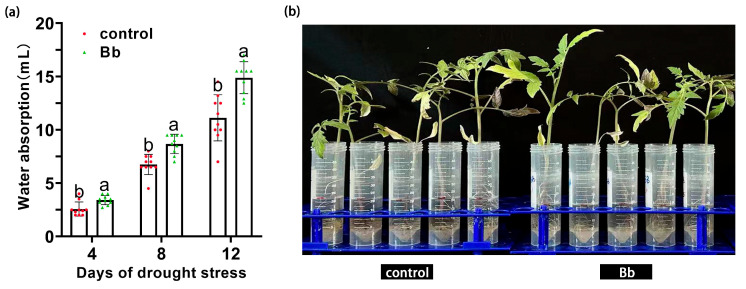
Effect of *B. bassiana* colonization on water absorption ability of tomato seedlings. (**a**) Effect of *B. bassiana* colonization on water absorption 4, 8, and 12 days after drought treatment, and (**b**) phenotypic effects of *B. bassiana* colonization on water absorption 12 days after drought treatment. Means represent average of all replications. *p* values indicate significant difference based on one-way ANOVA, respectively. Different letters above bars indicate significant differences between four treatments (*p* < 0.05).

**Figure 3 ijms-25-11949-f003:**
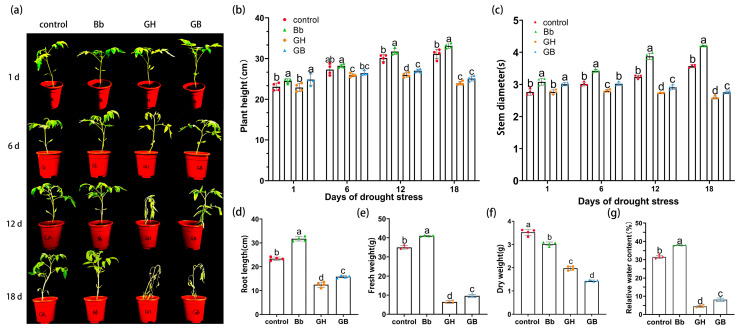
Effects of *B. bassiana* on the growth phenotypes, plant height, stem diameter, root length, fresh and dry weights, and relative water content of the tomato seedlings at different stages of drought stress. (**a**) Effects on overall growth after 1, 6, 12, and 18 days of drought stress, (**b**) Tomato seedling height, (**c**) stem diameter at different stages of drought stress, (**d**) root length, (**e**) fresh weight, (**f**) dry weight, and (**g**) relative water content on day 18 of drought stress. Control plants were root irrigated with 0.05% (*v*/*v*) Tween-80 root; Bb plants were root irrigated with *B. bassiana*; GH plants were treated with natural drought after root irrigation with 0.05% (*v*/*v*) Tween-80; GB plants were root irrigated with *B. bassiana* then subjected to natural drought treatment. Means represent the average of all replications. *p* values indicate a significant difference using one-way ANOVA, respectively. Different letters above the bars indicate significant differences between the four treatments (*p* < 0.05).

**Figure 4 ijms-25-11949-f004:**
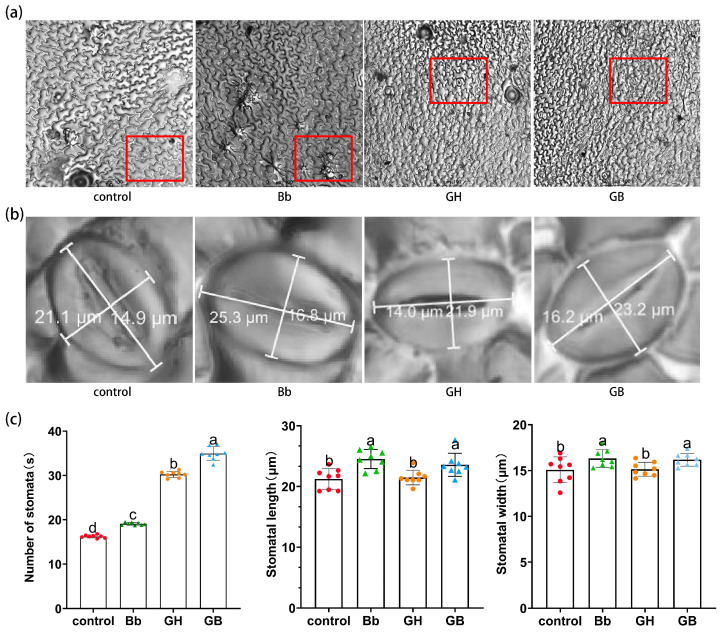
Changes in visual field with 40× field and stomatal length, width, and number in tomato leaves on day 12 of drought stress in each treatment. (**a**) Changes in stomatal number under microscope 20× field of view, (**b**) Changes in stomatal size under microscope 20× field of view, and (**c**) Effects of *B. bassiana* colonization on stomatal numbers, stomatal length, and stomatal width. The red box indicates that the number of stomata in the leaves of *B. bassiana* increased after colonization under drought stress (Bb vs. control, GB vs. GH). In Figure 4a,b, holes with convex lenticular shape and irregular distribution are stomata of tomato plants. Number, length, and width of stomata in tomato leaves treated with *B. bassiana* increased compared with those treated without *B. bassiana*. Means represent average of all replications. *p* values indicate significant difference using one-way ANOVA, respectively. Different letters above bars indicate significant differences between four treatments (*p* < 0.05).

**Figure 5 ijms-25-11949-f005:**
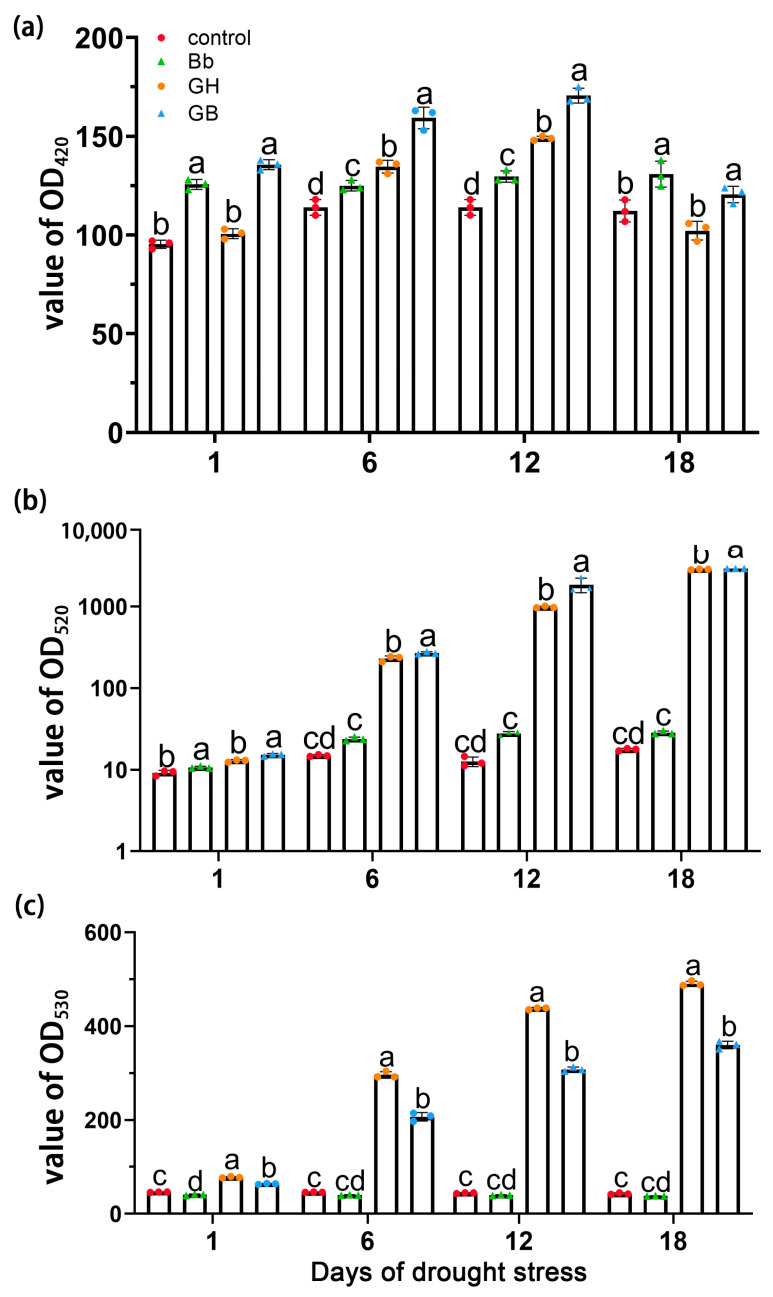
Effect of colonization of *B. bassiana* on (**a**) POD, (**b**) PRO, and (**c**) MDA activity in tomato seedlings at different stages of drought stress. Treatments are as described in Figure 2. Means represent average of all replications. *p* values indicate significant difference using ANOVA, respectively. Different letters above bars indicate significant differences between four treatments (*p* < 0.05).

**Figure 6 ijms-25-11949-f006:**
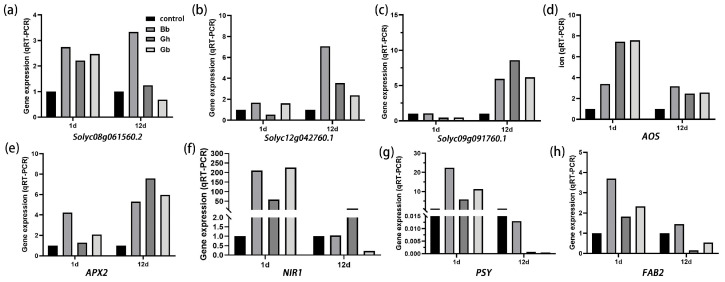
Expression levels of eight key genes related to stomatal development and drought resistance in tomato seedlings at different stages of drought stress. Treatments are as described in Figure 2. Values were determined using qRT-PCR. Expression levels of (**a**) *Solyc08g061560.2,* (**b**) *Solyc12g042760.1,* (**c**) *Solyc09g091760.1,* (**d**) *AOS,* (**e**) *APX2,* (**f**) *NIR1,* (**g**) *PSY,* and (**h**) *FAB2* of key genes related to stomatal development and drought resistance in tomato seedlings at different stages of drought stress.

**Figure 7 ijms-25-11949-f007:**
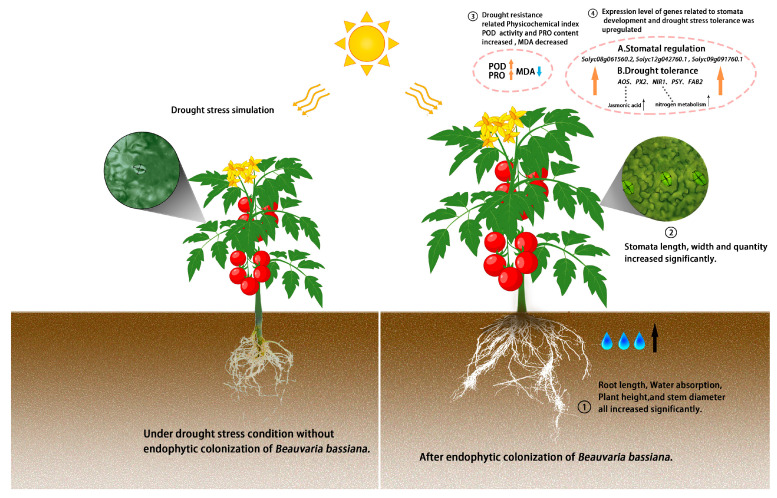
Schematic representation of effect of endophytic colonization of *B. bassiana* on drought resistance in tomato via “water spender” pathway. (The up arrow indicates that the gene is up-regulated and the down arrow indicates that the gene is down-regulated.)

## Data Availability

Data available on request from the corresponding authors.

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
