# Peer review of "Endophytic Colonization of Beauveria bassiana Enhances Drought Stress Tolerance in Tomato via “Water Spender” Pathway"

_ijms, 2024, doi:10.3390/ijms252211949_

Round 1

Reviewer 1 Report

Comments and Suggestions for Authors

Dear Editor

Many thanks for considering me as a potential reviewer for the article "Endophytic colonization of Beauveria bassiana enhances drought stress tolerance in tomato via the “water spender” pathway ". The article is undoubtedly well-structured, well-presented and well-written. However, I have several observations that should be considered before proceeding further.

My observations are as follows.

·       Lines-32 (a) Please cite the said information, (b), would be nice to add some statistics on economic loss because of drought stress (sue this the scope and importance of your will sounds).

·       Line-40, Would be nice if you add a few sentences like (my suggestions) ‘plants being sessile, however, have certain strategies to cope stresses….. Interestingly, plant-microbes interaction had played an important role in such conditions….

·       Throughout the manuscript, there is no space at the end of sentences and citations, please pay attention.

·       Line-78 Previous studies add one/two more examples…

·       Line79 Zea mays L. (Z. mays)?

·       Line-380, what is PEG…. Appear first time…

·       Line-98 Why Control is in capital letters?

·       Line-100 PEG stands for what?......appears the first time in results

·       Line-110 ‘From left to right are Control, Bb, GH, GB treatment’ this is confusing and unclear from the photos, please label each picture for more clarity.

·       Figure-2 (a and b) the legends should be bold for easy differentiation.

·       Statistical letters on Figures 2 and 3’s graphs are very small, please increase their sizes.

·       The authors claim that ‘compared control, significant increase and decrease occurs’, but how much and/or percent (%)?.  I would like to add/mention the percent increase and decrease….you can see the formula for the percentage effect calculation ‘DOI: 10.3390/metabo11110769.

·       Add a statistical analysis section at the end of Material and Methods.

·       Lines 154-164 ‘Similar to the results of plant height, significant differences in stem diameter were also observed between treatments on days 1, 6, 12, and 18 after drought stress (p<0.05; Over time, the stem diameter in the groups under drought stress was significantly lower than that of the control and Bb treatments. The stem diameter of seedlings treated with B. bassiana was higher than that of those treated without B. bassiana (Bb vs. Control and GB vs. GH) under identical watering conditions, respectively. On days 1, 6, 12 and 18 after drought stress, the stem diameter of seedlings in the GB treatment group was significantly higher than that of plants in the GH treatment group. Meanwhile, stem diameter was also significantly higher in the Bb treatment compared to the Control group. These findings indicate that irrigation with B. bassiana caused an increase in the stem diameter of tomato seedlings under both normal and drought stress conditions. How the authors can build these empirical statements? Where are the photos/results?....please clarify this!!

·       Figure 5. Why the same legends repeated in all graphs (a-c), I will suggest just in one, (b) also please remove the ‘days of drought of stress and 1, 6, 8 and 12 from graphs a and b, just leave it in graphs c. [check DOI: 10.3390/metabo11110769. (Figures 3 and 4)].

·       Please re-write your conclusion with more details, thereby adding and emphasizing each parameter positively and/or negatively increased/affected.

Comments on the Quality of English Language

Dear Editor/authors,

The article has interesting findings; however, some information is difficult to understand. I will suggest the authors consider the article for extensive English editing by a native English speaker and/or professional in the field.

Thanks

Author Response

Comments1: Lines-32 Please cite the said information, would be nice to add some statistics on economic loss because of drought stress (sue this the scope and importance of your will sounds).

Response 1: Thank you very much for your suggestion. The above information has been quoted in the revised manuscript to increase the relevant statistics of economic losses caused by drought.

Comments 2:Line-40, Would be nice if you add a few sentences like (my suggestions) 'plants beingsessile, however, have certain strategies to cope stresses..... Interestingly, plant-microbes interaction had played an important role in such conditions....

Response 2:  Thanks for the gentle advice. We have added this statement in the revised manuscript.

Comments 3:  There are some questions about the details, Such as: Line79 Zea mays L. (Z. mays)?Line-380, what is PEG.... Appear first time. Line-98 Why Control is in capital letters? Line-100 PEG stands for what?.....appears the first time in results; Line-110 'From left to right are Control, Bb, GH, GB treatment this is confusing and unclear from the photos, please label each picture for more clarity; Figure-2 (a and b) the legends should be bold for easy differentiation; Statistical letters on Figures 2 and 3's graphs are very small, please increase their size; Line79 Zea mays L.(Z. mays)? Figure 5. Why the same legends repeated in all graphs (a-c), l will suggest just inone, (b) also please remove the 'days of drought of stress and 1, 6, 8 and 12 fromgraphs a and b, just leave it in graphs c. [check DOl: 10.3390/metabo11110769(Figures 3 and 4)]. Throughout the manuscript, there is no space at the end of sentences and citationsplease pay attention.

Response 3: Thank you very much for reviewing this article so carefully and pointing out the above detailed defects. We have revised and improved these problems in the revised manuscript as following:

Changed the “Z. mays” to “Zea mays L.”; It is “polyethylene glycol” that the PEG appear for the first time; The “Control” has been changed to “control”; Labeled each picture to clarity Bb, GH, GB treatments; The legends have been bold for easy differentiation; Increased the size of statistical letters on Figures 2 and 3's graphs; The end of sentences and citations throughout the manuscript have been rechecked for the space.

Comments 4: The authors claim that 'compared control, significant increase and decrease occurs, but how much and for percent(%)? would like to add/mention the percent increaseand decrease....you can see the formula for the percentage effect calculation 'DOl: 10.3390/metabo11110769.

Response 4: Thank you very much for your suggestion. We have supplemented data in the revised draft according to the calculation formula of the relevant percentage increase in the references you provided.

Comments 5: Add a statistical analysis section at the end of Material and Methods。

Response 5: Thank you very much for your suggestion. I have added the statistical analysis section at the end of the materials and methods in the revised manuscript.

Comments 6: Lines 154-164 'Similar to the resuits of plant height, significant differences in stemdiameter were aiso observed between treatments on days 1,6, 12, and 18 afterdrought stress (p<0.05; Over time, the stem diameter in the groups under droughtstress was significantly lower than that of the control and Bb treatments. The stemdiameter of seediings treated with B. bassiana was higher than that of those treatedwithout B.bassiana(Bb vs.Control and GB vs. GH)under identical wateringconditions, respectively,On days 1,6,12 and 18 after drought stress, the stemdiameter of seedlings in the GB treatment group was significantly higher than that ofplants in the GH treatment group. Meanwhile, stem diameter was also significantlyhigher in the Bb treatment compared to the Control group. These findings indicatethat irrigation with B, bassiana caused an increase in the stem diameter of tomatoseedlings under both normal and drought stress conditions. How the authors canbuild these empirical statements? Where are the photos/results?....please clarify this!!

Response 6: Thank you for your question. These contents are described according to the results of Figure 3c, which has been supplemented at the end of the legends in this paragraph.

Comments 7: Please re-write your conclusion with more details, thereby adding and emphasizingeach parameter positively and/or negatively increased/affected.

Response 7: Thank you very much for your suggestion. We have added the details about significance in the conclusion of the revised manuscript, added and emphasized the positive impact of the relevant parameters of the study.

Reviewer 2 Report

Comments and Suggestions for Authors

The submitted manuscript addresses the current issue of using anti-stress agents to reduce the effects of water deficit on plants. Not many papers deal with that issue, the use of fungi, at least not in this context of measured parameters. This is where its originality lies. However, with respect to the use of PEG, the title of the manuscript is perhaps somewhat misleading, as it is not about drought. Please clarify the title. The manuscript is relatively carefully written, yet I recommend that it be completed or edited. The abstract gives a general description of the data obtained, but it might be useful to add specific values or differences, which would also increase its citability. The results are based on graphical representation of the measured values, but the quality of the graphs is lower, also considering its size. In the case of the photographs from the microscopic analysis, the description of the individual structures is missing. Not every reader is able to assess and describe the anatomical structure of plants. In the results it might be useful to add SPA analysis as well. The methodological procedures are adequate, but I recommend adding a tomato seed source. I believe that the species designation (line 360) is somewhat misleading, as it is probably a variety. I would also add the developmental stage according to BBCH within the age of the plants, as there may be growth variations within the age. In my opinion, the descriptions of the individual experiments are somewhat confusing. As I understand from the text, the plants were first grown in peat pots, then transferred to hydroponics with PEG and another part was still in soil. Please explain if this is indeed the case. In addition, it is important to note that PEG does not cause drought stress, but osmotic stress, which is physiologically different. Please recheck the discussion as well, it seems descriptive in places. Please recheck the citation of individual sources, it is not always consistent as journals are cited in full title and abbreviations.

Author Response

Comments 1: The submitted manuscript addresses the current issue of using anti-stress agents to reduce the effects of water deficit on plants. Not many papers deal with that issue, the use of fungi, at least not in this context of measured parameters. This is where its originality lies. However, with respect to the use of PEG, the title of the manuscript is perhaps somewhat misleading, as it is not about drought. Please clarify the title.

Response 1: Thank you very much for your gentle advice. PEG causes osmotic stress in plants, which has been used in previous studies to simulate drought stress. Therefore, we think that summarizing it as "drought stress" in the title is acceptable.

Comments 2: The manuscript is relatively carefully written, yet I recommend that it be completed or edited. The abstract gives a general description of the data obtained, but it might be useful to add specific values or differences, which would also increase its citability.

Response 2: Thank you for your meaningful suggestion. We have added a general description of the data.

Comments 3:The results are based on graphical representation of the measured values, but the quality of the graphs is lower, also considering its size.

Response 3: Thank you for your suggestion. We have increased the resolution of the image to 300 dpi.

Comments 4: In the case of the photographs from the microscopic analysis, the description of the individual structures is missing. Not every reader is able to assess and describe the anatomical structure of plants.

Response 4: Thank you for your meaningful suggestion. We highlight the number of stomata per unit area in Figure 4a to increase readers' understanding of the anatomical structure.

Comments 5:In the results it might be useful to add SPA analysis as well.

Response 5: Thank you for your gentle suggestion. Unfortunately, we do not understand SPA analysis. However, we believe that the existing data analysis is based on other relevant literature and can fully express the experimental results.

Comments 6: The methodological procedures are adequate, but I recommend adding a tomato seed source. I believe that the species designation (line 360) is somewhat misleading, as it is probably a variety.

Response 6: Indeed, the “BEAUTY” is a variety of tomato, which bred by Jilin Dalu Seed Industry Co., Ltd. 360 (Gongzhuling, Jilin Province, China).

Comments 7: I would also add the developmental stage according to BBCH within the age of the plants, as there may be growth variations within the age. In my opinion, the descriptions of the individual experiments are somewhat confusing.

Response 7:Thank you for your meaningful advice. All of our experiments were conducted during the tomato growth period of BBCH 30-39, which has been described in the methodology.

Comments 8: As I understand from the text, the plants were first grown in peat pots, then transferred to hydroponics with PEG and another part was still in soil. Please explain if this is indeed the case.

Response 8: In PEG experiment, the plants were first grown in peat pots, then transferred to hydroponics with PEG. While in the potted simulated drought experiment, all tomato plants were individually planted and treated.

Comments 9: In addition, it is important to note that PEG does not cause drought stress, but osmotic stress, which is physiologically different.

Response 9: As forementioned answer, PEG can cause osmotic stress, but it has been used in many studies to simulate drought stress.

Comments 10: Please recheck the discussion as well, it seems descriptive in places.

Response 10: Thanks for your gentle reminder. We have rechecked the discussion to reduce descriptive content.

Comments 11:Please recheck the citation of individual sources, it is not always consistent as journals are cited in full title and abbreviations.

Response 11: Thanks for your gentle reminder. We have rechecked the citation of individual sources to make which is consistent as journals are cited in full title and abbreviations.

Round 2

Reviewer 1 Report

Comments and Suggestions for Authors

Dear Editor,

I am thankful to the authors of accepting/correcting my suggestions in their article. Now the article is much more refined, however, I have noticed some minor corrections please do consider it, accordingly, in order to improve the quality of the article.

·       I ma not happy with the English language a number of issues are there in the text, especially, spacing, dots, spelling and grammatical issues, please do consider English editing,

·       Line-35 please correct it loss[1]. Please check throughout article,

·       Lines 43-45 please cite the newly added information,

·       Line-85 ‘Previous studies’ where is many citation? Please correct and/or add citations,

·       Lines 89-93 please add specific and more clearer objectives,

·       Figure-4 (cde) please add legends to one graph,

·       Figure 5, please add units i.e. MDA contents (unit) and so on…

·       Figure 7, add details and explain the numbers 1,2, 3…..in the image

Comments on the Quality of English Language

Dear Editor/authors,

The article is much more refine, however, there several technical issues with the Egnlish language. Would be nice to consider English editing. 

Thanks

Author Response

Comments 1: Line-35 please correct it loss[1]. Please check throughout article,

Response1: The contents about the loss caused by drought are included in reference 1. We have rechecked the references throughout the manuscript.

Comments 2: Lines 43-45 please cite the newly added information,

Response 2:  We have cited the reference ‘Yang, C., Han, N., Inoue, C., Yang, Y. L., Nojiri, H., Ho, Y. N., Chien, M. F. Rhizospheric plant-microbe synergistic interactions achieve efficient arsenic phytoextraction by Pteris vittata. J Hazard Mater. 2022, 434, 128870.’

Comments 3: Line-85 ‘Previous studies’ where is many citation? Please correct and/or add citations,

Response 3: Thanks for your advice. We have modified ‘Previous studies’ to ‘Previous study’.

Comments 4: Lines 89-93 please add specific and more clearer objectives,

Response 4: We have supplemented the specific objective in line 94, ‘However, few reports have examined the effect of endophytic colonization of B. bassiana on drought tolerance in tomatoes, as well as the mechanism and pathway which involved ’.

Comments 5: Figure-4 (cde) please add legends to one graph,

Response 5: Thanks for your gentle advice. We have added legends of Figure 4 (cde) to on graph, the corresponding Figure. 4 in the manuscript has been changed.

Comments 6: Figure 5, please add units i.e. MDA contents (unit) and so on…

Response 6: Thanks for your gentle advice. The units of manuscript in Figure 5 have been added.

Comments 7: Figure 7, add details and explain the numbers 1,2, 3…..in the image

Response 7: Thanks for your gentle advice. The corresponding Figure. 7 in the manuscript has been modified according to your advice.

Comments 8: There several technical issues with the Egnlish language. Would be nice to consider English editing.

Response 8: Thanks for your reminder. The editor has informed us that they will ask an English editor to help with our English revision.